# Preparation and Characterization of Silica-Based Ionogel Electrolytes and Their Application in Solid-State Lithium Batteries

**DOI:** 10.3390/polym15173505

**Published:** 2023-08-22

**Authors:** Ji-Cong Huang, Yui Whei Chen-Yang, Jiunn-Jer Hwang

**Affiliations:** 1Institute of Polymer Science and Engineering, National Taiwan University, Taipei 10617, Taiwan; d10549007@ntu.edu.tw; 2Department of Chemistry, Chung Yuan Christian University, Chung Li 32023, Taiwan; yuiwhei@cycu.edu.tw; 3Department of Chemical Engineering, Army Academy, Chung Li 32092, Taiwan; 4Center for General Education, Chung Yuan Christian University, Chung Li 32023, Taiwan

**Keywords:** ionogel, electrolytes, lithium battery, ionic conductivity, silica

## Abstract

In this study, tetraethyl orthosilicate (TEOS) and methyltriethoxysilane (MTES) were used as precursors for silica, combined with the ionic liquid [BMIM-ClO_4_]. Lithium perchlorate was added as the lithium-ion source, and formic acid was employed as a catalyst to synthesize silica ionogel electrolytes via the sol–gel method. FT-IR and NMR identified the self-prepared ionic liquid [BMIM-ClO_4_], and its electrochemical window was determined using linear sweep voltammetry (LSV). The properties of the prepared silica ionogel electrolytes were further investigated through FT-IR, DSC, and ^29^Si MAS NMR measurements, followed by electrochemical property measurements, including conductivity, electrochemical impedance spectroscopy (EIS), LSV, and charge–discharge tests. The experimental results showed that adding methyltriethoxysilane (MTES) enhanced the mechanical strength of the silica ionogel electrolyte, simplifying its preparation process. The prepared silica ionogel electrolyte exhibited a high ionic conductivity of 1.65 × 10^−3^ S/cm. In the LSV test, the silica ionogel electrolyte demonstrated high electrochemical stability, withstanding over 5 V without oxidative decomposition. Finally, during the discharge–charge test, the second-cycle capacity reached 108.7 mAh/g at a discharge–charge rate of 0.2 C and a temperature of 55 °C.

## 1. Introduction

With the advancement of technology, the demand for portable 3C products, such as smartphones, wearable devices, VR headsets, and laptops, is rapidly increasing. These devices are evolving to become lighter, thinner, shorter, and smaller. Consequently, there is a growing need for batteries with lighter weight, reduced thickness, compact size, and flexibility. Recently, the international community has dramatically emphasized carbon reduction, leading to the active development of electric vehicles in various countries. Lightweight and high-performance batteries are the key components of electric cars, making developing high-energy-density secondary batteries a popular research topic in recent years [1,2,3].

Currently, the most commonly used batteries in portable 3C products (such as computer, communication, and consumer electronic products) are polymer lithium-ion secondary batteries, which have a lower risk of leakage and explosion, greatly enhancing the safety of the battery. They also have superior thinness and flexibility compared to other secondary batteries. The main components of polymer lithium-ion secondary batteries include the cathode, anode, and polymer electrolyte membrane. However, developing a polymer electrolyte membrane that combines high ionic conductivity and separator functionality still presents challenges [4,5,6]. For example, there are difficulties in overcoming interface instability between the membrane and lithium metal or high-potential cathode materials, and including a large amount of organic solvents may reduce the membrane’s mechanical properties. As a result, many research teams are working together to develop solid polymer electrolytes (SPEs) with high room-temperature conductivity, high chemical stability, and excellent mechanical properties [7].

In recent years, ionogels have been widely used in various fields, such as electrolyte materials, gas separation, biocatalytic solvents, etc. [8,9]. The widespread attention ionogels have received is mainly due to their characteristics (with suitable choices of components): low volatility, low flammability, high ionic conductivity, and wide electrochemical windows. These features are critical for optimizing solid-state electrolytes in secondary lithium batteries. Neouze et al. [10] used a non-hydrolytic sol–gel method to prepare ionogels by immobilizing ionic liquids within a silica framework. These ionogels have both the mechanical properties of silica and the high ionic conductivity and thermal stability of ionic liquids. Lunstroot and Vioux et al. [11] incorporated europium (III) tetrakis β-diketonate into an imidazolium-based ionic liquid and encapsulated it within a silica network to prepare luminescent ionogels. These transparent organic–inorganic hybrid materials contain many ionic liquids (80 vol%) and exhibit high conductivity, producing intense and high-chroma red light under infrared radiation detection.

Ahmad et al. [12] used the sol–gel method with tetraethyl orthosilicate (TEOS) as a precursor and 1-ethyl-3-methylimidazolium bis(perfluoroethylsulfonyl)imide (EMI-PFSI) ionic liquid to prepare thin-film materials. These solid films exhibit high conductivity similar to that of liquid electrolytes, with conductivities up to 10^−2^ Scm^−1^ at 25 °C. These materials achieved excellent results when applied to electrochromic devices. Kim et al. [13] mixed ionic liquid [BMIM][TFSI] with solid acid CsHSO_4_ to prepare ionogels with high proton conductivity, thermal stability, and non-volatility. They pointed out the potential applications of these materials in fuel cells, water electrolysis, dehydrogenation, or electrochemical sensors. Mizumo et al. [14] mixed [TFSI] anion-containing ionic liquids with LiTFSI or LiBF_4_ to prepare ionogel electrolytes, proposing a novel concept for various energy electrolytes.

Echelmeyer et al. [15] developed a SiO_2_/[BMIM]BF_4_/LiTf ionogel electrolyte with suitable ionic conductivity and mechanical stability at room temperature. The results showed that due to the interaction between the ionic liquid and the silica framework, the ionogel electrolyte had good chemical flexibility and ionic conduction properties. Further research indicated that as long as the mechanical properties of the materials are improved, these materials can potentially become a new development trend for lithium battery electrolytes. Petit et al. [16] encapsulated lithium ions in SiO_2_/BMI-TFSI/LiTFSI ionogels and conducted rapid-field nuclear magnetic resonance tests to understand ion kinetics and explain the migration of anions and cations in ionic solutions. Gayet et al. [17] added polymers during the sol–gel process to create PMMA/SiO_2_/[BMIm][TFSI] hybrid ionogels. Adding a small amount of PMMA not only improved the mechanical properties of the ionogel but also increased the encapsulation of ionic liquids (up to 90%), addressing the issue of insufficient mechanical strength in ionogels. Vioux et al. [18] provided an essential review of ionogels, demonstrating their various functional applications. They selected different ionic liquids and encapsulation materials, indicating their potential value in multiple fields, with the energy sector being one of the most promising areas. Wu et al. [19] applied ionogels to lithium battery electrolytes and assembled them into batteries for charge–discharge testing. This marked the first documented instance of testing ionogel electrolytes in a half-cell configuration.

In a study by Bideau et al. [20], SiO_2_/PYR_13_-TFSI/LiTFSI solid ionogel electrolytes were used to replace conventional liquid electrolytes and separators. The electrolytes were directly prepared on the cathode LiNi_1/3_Mn_1/3_Co_1/3_O_2_ to improve the poor solid-electrode–electrolyte interface. Xie et al. [21] investigated the use of polymer PMMA, ionic liquid [Bmim][TFSI], and lanthanide material PtEu_2_ to produce cold light ionogels, revealing high thermal stability, cold light emission, and high ionic conductivity. They suggested that these ionogels were highly suitable for applications in electrochemistry and electrolytes.

MacFarlane et al. [22] prepared ionogels using ionic liquid [BMIM][BF_4_] and TEOS through the sol–gel method, studying the influence of different silica contents on thermal stability. The results showed that the thermal stability of these ionogels could reach up to 450 °C, indicating their potential as solid electrolytes. Bideau et al. [23] prepared ionogels using ionic liquid [Pyr_13_TFSI] and TMOS through the sol–gel method, examining the effects of different silica-to-ionic liquid ratios on ionogel microstructures. They found that smaller pores and better performance were achieved with higher silica content. M. J. Panzer et al. [24] used ionic liquids and PDMS to prepare ionogels via the sol–gel method. They discovered that the resulting ionogels exhibited good mechanical properties and ionic conductivity, with activation energies nearly identical to those of pure ionic liquids, suggesting their suitability as electrolyte materials. In summary, ionogels have attracted significant attention due to their unique properties and have been applied in various fields, especially in the energy sector. Developing ionogels with improved mechanical properties and ionic conductivity could lead to new advancements in lithium battery electrolytes and other energy-related applications.

This study used the sol–gel method to immobilize ionic liquids within a silica framework, creating a class of solid electrolytes known as ionogels. Although the literature [25] has indicated that ionogels possess high ionic conductivity and thermal stability, their application as lithium battery electrolytes still requires addressing the issue of insufficient mechanical properties for thin films. Therefore, this research combines ionogels and electrolytes by adding lithium salt to the ionogels (referred to as composite ionogel electrolytes) and introducing silicon precursors with methyl groups (methyltriethoxysilane, MTES) to enhance mechanical strength. This approach is expected to improve the inadequate mechanical strength of ionogels and facilitate their practical application in lithium batteries.

## 2. Experimental Section

### 2.1. Chemicals and Materials

Methylimidazole (C_4_H_6_N_2_) (ACROS Organics, Geel, Belgium); 1-Chlorobutane (C_4_H_9_Cl) (Sigma Aldrich, St. Louis, MO, USA); Sodium perchlorate (NaClO_4_) (Sigma Aldrich, St. Louis, MO, USA); Acetonitrile (CH_3_CN) (TEDIA, Fairfield, OH, USA); Lithium perchlorate (LiClO_4_) (ACROS Organics, Geel, Belgium); Methanol (CH_3_OH, MeOH) (99.5%, Juming Chemicals, Taipei, Taiwan); Formic acid (HCOOH, FA) (96%, TEDIA, Fairfield, OH, USA); Tetraethyl orthosilicate (TEOS, Si(OC_2_H_5_)_4_) (reagent grade, 98%, ACROS Organics, Geel, Belgium); Methyltriethoxysilane (MTES, CH_3_Si(OC_2_H_5_)_3_) (reagent grade, 98%, ACROS Organics, Geel, Belgium); Lithium metal (Li) (99.9%, Xuneng Ltd., Shenzhen, China). Super P (high conductivity carbon black) (UBIQ Technology Co. Ltd., Taipei, Taiwan); Poly(1,1-difluoroethylene) (PVDF) (UBIQ Technology Co., Taipei, Taiwan); 1-methyl-2-pyrrolidone (NMP, C_5_H_9_NO) (99%, Thermo Fisher Scientific Inc., Waltham, MA, USA).

### 2.2. Instrumentation

Dry box (Model: UNI-LAB, Mbraun, Garching, Germany); Fourier Transform Infrared Spectrometer (Model: FTS-7, BIO-RAD, Hercules, CA, USA); Differential scanning calorimetry (Model: Q-10, TA Instruments, New Castle, DE, USA); A.C. Impedance (Model: HP4192A, HP, Palo Alto, CA, USA); Potentiostat/galvanostat instrument (Model: AUTOLAB PGSTAT30, Metrohm Autolab, Utrecht, The Netherlands); Charge–discharge system (Model: LAND CT2100A, Landt Instruments, Wuhan, China); Nuclear magnetic resonance (NMR) (Model: JNM-ECZS 400MHz, JEOL, Tokyo, Japan); Scanning electron microscope (SEM) (Model: JSM-7600F, JEOL Ltd., Tokyo, Japan).

### 2.3. Preparation of Ionic Liquids (IL)

First, prepare BMIC (1-butyl-3-methylimidazole chloride) with the starting material being 1-methylimidazole. Under a nitrogen atmosphere, add 4 equivalents of 1-chlorobutane to the 1-methylimidazole. Use acetonitrile (CH_3_CN) as the solvent and maintain the temperature at around 70 °C. Stir the mixture in the dark for 7 days. After the reaction is complete, pour the solution into a separatory funnel and let it stand to form layers. The upper layer comprises excess 1-chlorobutane, while the lower layer is the BMIC-containing cyanide alkane solution. Collect the lower layer in a concentration flask and use a vacuum distillation apparatus to remove the 1-chlorobutane and acetonitrile. Then, use a freeze dryer to remove any remaining impurities to obtain a pale yellow BMIC.

To prepare the BMIM-ClO_4_ ionic liquid, dissolve the synthesized BMIC (1-butyl-3-methylimidazolium chloride) in a predetermined amount of sodium perchlorate (NaClO_4_, with a 1:1 molar ratio). Use acetone as the solvent and mechanically stir the mixture at room temperature for 7 days. At this point, BMIM-ClO_4_ ionic liquid and sodium chloride precipitate will form. Use vacuum filtration to remove the sodium chloride precipitate and a vacuum concentrator to remove the acetone. Finally, use a freeze dryer to remove any remaining impurities to obtain the BMIM-ClO_4_(1-butyl-3-methylimidazole perchlorate) ionic liquid with the chloride ions replaced by perchlorate ions. Next, purify the obtained BMIM-ClO_4_ ionic liquid using activated carbon. Mix the BMIM-ClO_4_ with activated carbon (activated carbon weight = BMIM-ClO_4_ weight × 0.2) and use ethanol as the solvent. Maintain the temperature at 70 °C and purify the mixture for 7 days. Then, use vacuum filtration to remove the activated carbon and a vacuum concentrator to remove the ethanol. Finally, use a freeze dryer to remove any remaining impurities to obtain the purified BMIM-ClO_4_ ionic liquid.

### 2.4. Preparation of Silica Ionogel Electrolytes

Tetraethyl orthosilicate (TEOS) and Methyltriethoxysilane (MTES) are silica precursors. Ionic liquid BMIM-ClO_4_ and lithium perchlorate (LiClO_4_) are added as the source of lithium ions, with formic acid added to assist the hydrolysis-condensation reaction. First, prepare solutions A and B. Solution A is a mixture of TEOS, MTES, and formic acid, allowing pre-hydrolysis of TEOS and MTES. Solution B is a mixture of ionic liquid BMIM-ClO_4_ and lithium perchlorate LiClO_4_. Next, mix solutions A and B, stirring to ensure homogeneous mixing. Use a micropipette to transfer a measured amount of the mixture into a mold. After placing it at 50 °C for 2 days to form a film, dry it in a vacuum oven at 115 °C for one day to create a silica ionogel electrolyte (SIE). The experimental procedure is shown in Figure 1. Due to the gel-like consistency of BMIM-ClO_4_, the silica ionogel electrolyte (SIE) obtained from its hydrolytic condensation with TEOS exhibits suboptimal mechanical strength. When fabricated into thin films, this results in surface cracking. To improve its processability and yield a film with enhanced mechanical properties, the proportion of MTES can be increased. For instance, with a molar ratio 1.0, the design is based on MTES/TEOS with 0.5 mole of MTES and 0.5 mole of TEOS. This leads to the formulation of a silica composite substrate. In this study, the substrate formulations were controlled at molar ratios of 0.2, 0.4, 0.6, 0.8, and 1.0. We synthesized a series of SiO_2_ ionogel polymer electrolytes and conducted identification and property analyses. The formulation and code of each sample are detailed in Table 1.

### 2.5. Material Identification and Analysis Methods

In this study, Fourier-transform infrared (FTIR) spectroscopy was utilized to identify the characteristic peaks of ionic liquids. Experimental conditions included taking an appropriate amount of the sample and blending it with KBr powder in a 1:100 ratio. This mixture was then finely ground and pressed to form a KBr pellet for spectroscopic analysis. The spectral resolution was 4 cm^−1^ with a scanning wavenumber range of 4000 to 400 cm^−1^. Differential scanning calorimetry (DSC) was employed to determine the glass transition temperature (Tg) of the silica-based ionogel electrolyte (SIE). This measurement provided insights into the relationship of its ionic conduction within a specific temperature range. Samples were evaluated with a mass ranging between 5 and 10 mg. Experiments were executed under a nitrogen atmosphere. The initial heating profile spanned from −120 °C to 120 °C at a controlled ramp rate of 10 °C/min. Following this, the sample underwent a cooling phase, descending at 10 °C/min until reaching −120 °C. Subsequently, a secondary heating cycle was administered, elevating the temperature from −120 °C to 30 °C at the same rate of 10 °C/min. During this second thermal excursion, the material’s glass transition temperature (Tg) was closely monitored and documented.

^1^H-NMR spectra of the as-prepared BMIM-ClO_4_ were characterized by a Bruker Avance Neo 9.4 Tesla NMR spectrometer (corresponding to a proton resonance frequency of approximately 400.13 MHz) equipped with a liquid iProbe. The sample was loaded into a single tube for analysis, utilizing D_2_O as the deuterated solvent. Trace amounts of H_2_O, resonating at δ = 4.7 ppm, were employed as the internal reference. The measurements were conducted at a maintained temperature of 25 °C. Analysis of the SIE using ^29^Si-NMR spectra was conducted with a Bruker 400 WB solid-state NMR spectrometer, operating at a proton frequency of 400.13 MHz. The NMR spectra were acquired under the following conditions: ^29^Si frequency of 79.48 MHz, a π/2 pulse of 3.9 µs. The single pulse sequence included a π/4 pulse of 3.9 µs, a decoupling duration of 6.3 µs, and a recycle delay of 100 s, totaling 2000 scans. Samples were packed into 4 mm zirconia rotors and spun at 9 kHz under an airflow. Octamethylcyclotetrasiloxane (Q8M8) was used as external secondary reference. The measurements were conducted at a maintained temperature of 25 °C.

### 2.6. Descriptions of Electrochemical Measurements

Ionic conductivities of silica ionogel electrolytes were evaluated between 30 °C and 100 °C using an HP4192A impedance analyzer (Agilent, Santa Clara, CA, USA) spanning frequencies of 0.1 Hz to 1 MHz. Measurements occurred in a vacuum chamber maintained at a pressure of 5 × 10^−2^ torr. The ionogels, with a diameter of 15 mm and a thickness of 100 μm, were sandwiched between two Au electrodes, each measuring 12 mm in diameter. All measurements employed a 50 mV rms amplitude setting, and no separators were utilized in the setup.

The LSV experiments were conducted utilizing a CR2032 coin cell. Within this cell, stainless steel, with a diameter of 12 mm, served as the working electrode, while metallic lithium of the same diameter acted as the counter electrode. Positioned between these electrodes was a silica ionogel electrolyte with a diameter of 15 mm and a controlled thickness of 100 μm. Notably, no additional separators were employed in this half-cell configuration. The assembly of the half-cell was meticulously carried out within a dry box (Model: UNI-LAB) to ensure that the humidity and oxygen levels remained below 1 ppm. Measurements were facilitated by a Potentiostat/galvanostatic instrument (Model: AUTOLAB PGSTAT30), with a voltammetric scan range set from 0 to 7 V and a scan rate of 5 mV/s, under a controlled temperature of 30 °C. A detailed examination was undertaken to observe the current response as a function of voltage variations.

Discharge–charge experiments utilized metallic lithium as the anode and LiFePO_4_ as the cathode (with a theoretical capacity of 170 mAh/g). A 1.0 g amount of anhydrous Super P (a highly conductive carbon black) and 1.0 g of a 4.0 wt% PVDF solution in NMP were combined in a 50 mL sample vial. This mixture was stirred at 200 rpm for an hour to ensure uniform blending. Subsequently, 8.0 g of LiFePO_4_ was added and stirred at 200 rpm for another hour. An additional 10 mL of NMP was incorporated upon achieving homogeneity, and the resulting slurry was stirred at 500 rpm for 4 h. This uniform paste was then blade-coated onto an aluminum foil substrate (paste thickness: 300 μm; foil thickness: 20 μm). The coated foil was dried overnight at 80 °C under vacuum conditions. Once dried, the electrode material was punched into 13 mm diameter discs. The final weight of each electrode disc was measured with an electronic balance to determine the loading of LiFePO_4_. The ionogel electrolyte prepared in this study was sandwiched between the anode and cathode, with a diameter of 15 mm and thickness maintained at 100 μm. Notably, no additional separators were employed within the cell. The assembly of the CR2032 lithium half-cell was conducted inside a dry box. Discharge and charge tests were executed using a charge–discharge system (Model: LAND CT2100A) at a rate of 0.2 C. The cells were positioned in a 55 °C environment to evaluate their charge–discharge performance.

## 3. Results and Discussion

### 3.1. Identification of BMIC and BMIM-ClO_4_ Ionic Liquid

Infrared spectroscopy is a technique that involves irradiating a sample with infrared light of a specific wavelength. Suppose the frequency of the infrared light satisfies the transition conditions of certain functional groups in the sample’s molecules. In that case, the molecules will absorb the infrared radiation energy at that wavelength and transition from a lower ground state vibrational energy level to a higher excited state vibrational energy level. An infrared absorption spectrum of the sample can be obtained by detecting the sample molecules’ absorption intensity at different wavelengths of infrared light. This study used infrared spectroscopy to identify the characteristic peaks of the ionic liquid and silica. The operating conditions were as follows: mix an appropriate amount of the sample with KBr powder in a 1:100 ratio, grind, and press into a KBr salt pellet. The resolution is 4 cm^−1^, and the scanning wavelength range is 4000~400 cm^−1^. Figure 1 shows the FTIR spectra of BMIC and BMIM-ClO_4_. From the figure, it can be seen that the characteristic absorption peaks for the C-H stretching vibration on the imidazole ring are located at 3158 cm^−1^ and 3119 cm^−1^; the characteristic absorption peaks for the C-H stretching vibrations of the -CH_3_ and -CH_2_ side chains on the imidazole ring are located at 2964 cm^−1^ and 2938 cm^−1^; the characteristic absorption peak for the -C=N stretching vibration on the imidazole ring is at 1574 cm^−1^; the characteristic absorption peak for the -CH_2_ bending vibration is at 1467 cm^−1^; and the absorption peak for the C-H bending vibration on the imidazole ring plane is at 1170 cm^−1^.

Furthermore, the most significant differences between the BMIM-ClO_4_ and BMIC spectra are the strong absorption peaks at 1097 cm^−1^ and 625 cm^−1^, which correspond to the stretching vibrations of ClO_4_^−^. By comparing the absorption spectra of the peaks mentioned above with the spectra in the literature, it can be confirmed that the synthesized ionic liquid is indeed pure BMIM-ClO_4_.

Figure 2 shows the ^1^H NMR spectrum of the ionic liquid. The figure shows that the chemical shift at δ = 0.83 is a triplet with an integration value corresponding to three hydrogen atoms, which is determined to be the chemical shift of the methyl group on the butyl side chain of the imidazole ring. At δ = 1.22, there is a sextet with an integration value of two hydrogen atoms, determined to be the chemical shift of the CH_2_ group adjacent to the methyl group on the butyl side chain of the imidazole ring. At δ = 1.76, there is a quintet with an integration value of two hydrogen atoms, which is determined to be the chemical shift of the middle CH_2_ group in the -CH_2_CH_2_CH_2_CH_3_ butyl side chain on the imidazole ring. At δ = 3.80, there is a singlet with an integration value of three hydrogen atoms, which is determined to be the chemical shift of the methyl group on the side chain of the imidazole ring. At δ = 4.10, there is a triplet with an integration value of two hydrogen atoms, which is determined to be the chemical shift of the first CH_2_ group in the -CH_2_CH_2_CH_2_CH_3_ butyl side chain on the imidazole ring. At δ = 7.33, 7.38, and 8.59, there are singlets with an integration value of one hydrogen atom each, determined to be the chemical shifts of the C-H groups on the imidazole ring.

Combining the FTIR and NMR results, we can conclude that the ionic liquid BMIM-ClO_4_ has likely been synthesized. Furthermore, according to the NMR chemical shifts reported in the literature: 0.945 (t, 3H), 1.377 (m, 2H), 1.928 (m, 2H), 4.055 (s, 3H), 4.362 (t, 2H), 7.712 (s, 1H), 7.769 (s, 1H), 9.021 (s, 1H) ppm, the successful synthesis of BMIM-ClO_4_ can be confirmed once again [26].

### 3.2. Identification and Analysis of Silica Ionogel Electrolyte

#### 3.2.1. Fourier Transform Infrared Spectroscopy (FTIR) for Detection and Analysis

The prepared silica ionogel electrolyte was analyzed using Fourier transform infrared spectroscopy (FTIR). As shown in Figure 3, the presence of -O-H bond absorption peaks can be observed at 3200~3600 cm^−1^ and 1620~1640 cm^−1^. The absorption peak at 3200–3600 cm^−1^ is mainly due to the O-H bond vibration of ≡Si-OH. The absorption peak at 930–950 cm^−1^ corresponds to the bond vibration of ≡Si-OH. Additionally, the absorption peak at 1000–1200 cm^−1^ is attributed to the stretching vibration of the ≡Si-O-Si≡ bond. These absorption peaks indicate the successful formation of a silica skeleton through the hydrolysis and condensation of tetraethyl orthosilicate (TEOS) and methyltriethoxysilane (MTES).

Furthermore, as shown in Figure 4, the addition of MTES does not cause any shift in this absorption peak, suggesting that the inclusion of MTES does not affect the structure of the silica skeleton. The absorption peak at 1275 cm^−1^ corresponds to the C-H deformation vibration of the Si-R bond in silane. Figure 4 shows that the intensity of this absorption peak increases with the addition of MTES. Moreover, the characteristic absorption peaks of the ionic liquid can be observed in the spectrum: at 3100–3200 cm^−1^ for the -CH stretching of the imidazole ring; at 2800–3000 cm^−1^ for the aliphatic -CH stretching; at 1500–1600 cm^−1^ for the C=N stretching of 1-methylimidazole; at 1430–1470 cm^−1^ for the aliphatic -CH_2_ bending; and at 626–640 cm^−1^ for the ClO_4_^−^ characteristic absorption peak. These characteristic absorption peaks confirm that the ionic liquid has been successfully incorporated into the material.

#### 3.2.2. Detection and Analysis of Glass Transition Temperature (Tg) Using Differential Scanning Calorimeter (DSC)

For all solid electrolytes, the Tg is an important indicator, reflecting the ease of ion conduction, which is related to the conductivity of the electrolyte. Therefore, we will investigate the influence of pure ionic liquid and scaffold structure on the DSC thermal curve. From Figure 5, we can see that the Tg of the pure ionic liquid is −72.2 °C. In the thermal profile of BMIM-ClO_4_, besides the observed Tg at −72.2 °C, an exothermic peak (Tc) emerged at −40.5 °C. It is postulated that BMIM-ClO_4_, having undergone a comprehensive heating and cooling cycle, exhibits a more ordered molecular arrangement. Strong electrostatic interactions might be present within BMIM-ClO_4_, leading to certain crystallization events, thus accounting for the appearance of the exothermic peak. As presented in Table 2, it was observed that the Tg changed after adding lithium perchlorate. The Tg increases to −64.2 °C due to the increase in viscosity. The prepared samples (SIEM0~SIEM10) all show the Tg of the ionic liquid. The Tg of SIEM0 (−62.8 °C) is slightly higher than that of BMIM-ClO_4_ + LiClO_4_ due to the interaction between Li^+^ and oxygen atoms on the silica scaffold. When MTES is added, the number of oxygen atoms on some silica surfaces decreases (SIEM2), weakening the attraction between Li^+^ and oxygen atoms, leading to a decrease in the Tg (−80.8 °C). However, when the methyl group continues to increase (SIEM4), according to the literature [27], as the methyl group is a hydrophobic functional group, phase separation occurs when combined with TEOS, as shown in Figure 6, causing the Tg to rise again. When there is an excess of methyl groups (SIEM10), they may aggregate, enabling oxygen atoms to interact with Li^+^ again, resulting in a slight upward trend in the Tg of SIEM10 (−62.8 °C). From SIEM0 to SIEM10, a slight shift in the Tg can be observed, confirming that the ionic liquid is indeed encapsulated in the silica scaffold, referred to as the confinement effect.

“Confinement effects” refer to the phenomena wherein a substance’s physical, chemical, or biological characteristics (such as molecules, ions, particles, etc.) exhibit behaviors different from those in an unrestricted environment when confined within a relatively small space or volume. Such effects are prevalent in nanostructures, thin films, and pores on the micro or nanometric scale. Due to this spatial confinement, the material’s properties are influenced by its environment, leading to potential changes in its kinetics, electrochemistry, thermal properties, and mechanical attributes [28,29].

The confinement effects in silica ionogel electrolytes focus on the property changes of BMIM-ClO_4_ when spatially restricted within a silica scaffold. Ion liquids confined within a networked structure may exhibit alterations in their physical properties, such as their glass transition temperature (Tg), due to the constrained mobility of the ionic liquid. Indeed, there is a significant decrease when comparing the Tg of SIEM2 (−80.8 °C) with that of BMIM-ClO_4_+ LiClO_4_ (−64.2 °C). The Tg variations for other ionogel electrolytes might be minor, hypothesizing that a more stable phase equilibrium state is established between the silica networked structure and the ionic liquid [30]. However, from the SEM observations of the ionogel electrolyte, we can ascertain that the ionic liquid is indeed encapsulated within the silica scaffold.

#### 3.2.3. Identification by Solid-State ^29^Si MAS NMR Spectroscopy

To understand the bonding situation in the cross-linked network structure formed during the sol–gel reaction, one must use solid-state ^29^Si MAS NMR spectroscopy for identification. According to the literature [31], if the cross-linked network structure is Q^4^, characteristic absorption peaks will occur at −107 to −111 ppm; if it is Q^3^, characteristic absorption peaks will occur at −99 to −102 ppm; and if it is Q^2^, characteristic absorption peaks will occur at −91 to −93 ppm. The silicon dioxide ionogel electrolyte prepared in this study contains methyl to have characteristic absorption peaks of T^3^ and T^2^. If the cross-linked network structure is T3, characteristic absorption peaks will occur at −64 to −67 ppm. If it is T^2^, characteristic absorption peaks will occur at −55 to −57 ppm.

Once the proportions of each absorption peak are known, the degree of condensation (Dc) can be calculated using the following formula. The higher the degree of condensation, the more complete the hydrolysis condensation. Figure 7 and Table 3 show that the degree of condensation for all five samples, from SIEM2 to SIEM10, is over 80%, indicating that the degree of hydrolysis condensation tends towards completeness.
$$D_{c}=([\frac{4Q^{4}+3Q^{3}+2Q^{2}}{4}]+[\frac{3T^{3}+2T^{2}}{3}])\times 100\%$$

#### 3.2.4. Ionic Conductivity Analysis of Ionogel Electrolytes

The measurement method for ionic conductivity involves first measuring the impedance value and then calculating it using the ionic conductivity formula: σ = L/(R × A), where L is the thickness (cm) of the polymer ionogel electrolyte, R is the impedance value (ohm), and A is the conduction area (cm^2^) of the silica ionogel electrolyte. Many factors affect ionic conductivity, such as the interaction between silica and lithium ions, lithium salt concentration, and temperature. Table 3 shows the room temperature (30 °C) ionic conductivity of the ionic liquid (BMIM-ClO_4_) and silica ionogel electrolytes (SIEM2, SIEM4, SIEM6, SIEM8, SIEM10).

As can be seen from Table 3, the room temperature ionic conductivity of the pure ionic liquid is 1.46 × 10^−3^ S/cm. After adding silica, the ion conduction becomes less mobile in the ionic liquid, decreasing conductivity. Although the conductivity of SIEM4 is higher than that of SIEM2, the difference between the two is insignificant, and their conductivities are nearly identical. The conductivity of SIEM6 and SIEM8 is significantly lower, while the conductivity of SIEM10 increases. The reason for this, as mentioned earlier, is that the substitution of methyl groups initially reduces the number of oxygen atoms, decreasing the chance for oxygen atoms to interact with Li^+^. However, when more methyl groups are present, they may aggregate, allowing oxygen atoms to interact with lithium ions again, resulting in increased conductivity.

#### 3.2.5. The Relationship between Ionic Conductivity and Temperature

Figure 8 shows the temperature-dependent conductivity plot for ionic liquid and silica ionogel electrolytes. We obtain the Arrhenius plot by taking the logarithm (Log σ) of the measured conductivity and plotting it against 1000/T. The plot shows that the changes are close to linear, indicating that the system follows the Arrhenius equation within the range of 30 °C to 100 °C:σ = σ_0_exp(−Ea/RT)(1)
where σ is the conductivity, Ea is the activation energy, R is the gas constant (=8.314 J K^−1^ mol^−1^), and σ_0_ is the pre-exponential factor. The activation energy Ea can be obtained from the slope m of the straight line in the Arrhenius plot: Ea = −2.303 Rm.

Table 4 presents the activation energies of the ionic liquid (BMIM-ClO_4_) and silica ionogel electrolytes. From the Table 4, it can be observed that the activation energy of the pure ionic liquid is 28.9 kJ/mol. After adding a small amount of silica, the activation energy decreases to 17.8 kJ/mol, indicating that adding silica provides an alternative conduction pathway for ions, reducing the activation energy. However, as the proportion of silica increases (SIEM4, SIEM6, and SIEM8), the activation energy also rises, consistent with the trend observed in ionic conductivity. Within this temperature range, the conductivity of silica ionogel electrolytes follows the Arrhenius equation. This suggests a significant correlation between the thermal properties and ionic conductivity of the series of silica ionogel electrolytes synthesized in this study and their activation energy (Ea).

#### 3.2.6. Linear Scanning Voltammetry (LSV) Measurement of Ionogel Electrolytes

Linear sweep voltammetry (LSV) is a crucial technique in electrochemical analysis employed to investigate the electrochemical behavior of electrode processes. During this process, the electrode’s potential is linearly changed (swept) with time, thereby measuring the current response associated with potential variations. An LSV experiment involves applying a controlled voltage to the electrode, which is changed linearly. As the voltage changes, the current at the electrode is also altered. This fluctuation in current provides us with insights into the oxidation and reduction reactions in the electrolyte. When a specific potential triggers a conspicuous change in current, we can infer that an oxidation or reduction reaction occurs at this potential. This potential is termed the oxidation or reduction potential, depending on the reaction type. A vital application of this method is to ascertain the stable voltage range of the electrolyte. When a current undergoes abrupt changes, it can be inferred that the electrolyte starts to undergo oxidative or reductive decomposition. Thus, this technique can assist researchers in understanding the chemical stability of the electrolyte and the highest and lowest voltages it can endure. When the voltage increases to a certain degree, the current rises sharply, indicating that the electrolyte material begins oxidative decomposition at that voltage. Therefore, LSV experiments can provide information on the stability of the electrolyte under specific voltage conditions [32,33].

In this study, we use LSV to explore the maximum voltage our prepared silica ionogel electrolyte can withstand without causing oxidative decomposition. The application of this method aids us in gaining further insights into the performance and stability of the electrolyte under high-voltage conditions. We designed a CR2032 button cell to conduct this experiment. This cell comprises stainless steel as the working electrode, lithium metal as the counter electrode, and our prepared silica ionogel electrolyte is placed in between. The LSV experiment sets the voltammetric scanning range from 0 to 7 V and a scanning rate of 5 mV/s. This setup allows detailed observation of the current response to voltage variations. Figure 9 depicts a linear sweep voltammogram of the silica ionogel electrolyte. As can be seen from the figure, the electrolytes from SIEM2 to SIEM10 all exhibit a sharp rise in current only upon reaching a voltage of 5 V, indicating that our prepared silica ionogel electrolyte has an oxidative potential above 5 V. In other words, our prepared silica ionogel electrolyte demonstrates excellent high-voltage resistance in the designed lithium battery test and does not undergo oxidative decomposition under conditions below 5 V. This result has significant implications for enhancing lithium battery performance and stability.

#### 3.2.7. Electrochemical Impedance Spectroscopy (EIS) of Ionogel Electrolytes

Electrochemical impedance spectroscopy (EIS) is a technique widely utilized in electrochemical systems, providing information about the kinetics and mechanisms of electrochemical processes. EIS involves applying a small alternating current or voltage signal to an electrochemical system, then measuring the corresponding current or voltage response across a range of different frequencies. These impedance values are recorded and can be visualized in an impedance spectrum. An impedance spectrum graphically represents the real and imaginary parts (or phase and magnitude) of impedance as a function of frequency. Specific features in the EIS plot include the ohmic resistance (R_b_), composed of the resistances of the electrolyte, electrodes, and connectors, typically appearing at the far left of the spectrum; the charge transfer resistance (R_i_), which is the impedance to the transfer of electrons across the interface of the electrolyte and electrode, often represented as a semi-circle; and Warburg Impedance, which is the impedance due to the diffusion of ions within the electrode, typically appearing as a 45-degree line in the mid-frequency and low-frequency ranges [34,35].

EIS can help us understand various phenomena occurring in the silicon dioxide ionogel electrolyte during electrochemical processes, including charge transfer, diffusion, and interfacial phenomena. For this reason, we carried out EIS testing on a CR2032 button cell at 55 °C. This battery employs lithium iron phosphate (LiFePO_4_) as the positive electrode material and lithium metal as the negative electrode. As shown in Figure 10, the first point on the far left in the EIS represents the electrolyte’s inherent impedance (R_b_). The size of the semi-circle reflects the interfacial impedance (R_i_) between the electrode and the electrolyte, and the final sloping line represents the Warburg impedance, symbolizing the diffusion impedance of lithium ions within the electrode material. The associated impedance values have been compiled in Table 4 for reference.

Interfacial impedance is a crucial factor influencing the charge–discharge performance of lithium batteries. If the interfacial impedance is too high, it will be difficult for lithium ions to successfully de-intercalate from one electrode, traverse the electrolyte, and intercalate into the other electrode. Figure 10 and Table 4 show that SIEM2 has the lowest interfacial impedance. As the methyl content increases, the interfacial impedance also rises. This is due to the surface becoming rougher as the methyl content increases, as observed in the SEM and AFM surface morphology image in Figure 11. Thus, when the silicon dioxide ionogel electrolyte comes into contact with the electrode, the interfacial impedance correspondingly increases.

### 3.3. Charge and Discharge Test of Solid-State Lithium Batteries

Lithium-ion secondary batteries are the primary power source for many modern electronic devices. This type of battery has many advantages, including high energy density, a longer lifespan, and a relatively lower self-discharge rate. However, the electrolyte plays an essential role in lithium-ion batteries. It not only allows lithium ions to move between the cathode and anode during the charge and discharge process but also can affect the overall performance and lifespan of the battery. Therefore, the selection and development of the electrolyte is a critical area in the research on lithium-ion batteries.

In this study, the ionogel electrolytes underwent a vacuum freeze-drying process to ensure the complete removal of water content. The presence of water not only obstructs the intended reversible reactions within the lithium battery but could also significantly influence the material’s conductivity and ionic transport, subsequently diminishing the battery’s efficiency [36,37]. Since BMIM-ClO_4_ is inherently hydrophilic and the structure of silica also exhibits hydrophilic tendencies, all components and materials for lithium battery assembly underwent rigorous dehydration. They were stored in a dry box until fully assembled into coin cells, only after which subsequent battery charge–discharge tests were conducted. To understand the performance of silicon dioxide ionogel electrolytes applied in lithium batteries, we conducted tests with CR2032 lithium half batteries composed of the prepared silica ionogel electrolytes. The anode used was lithium metal, while the cathode selected was LiFePO_4_. Discharge and charge tests were conducted at a rate of 0.2 C, and the batteries were placed under a high-temperature environment (55 °C) to test their charge and discharge performance. Upon evaluation, we found the interface impedance of SIEM8 and SIEM10 too high, resulting in poor discharging and charging effects. Consequently, we selected SIEM2, SIEM4, and SIEM6 for further discharging and charging tests. Figure 12 presents the discharge–charge curves after the first discharge and charge cycles of SIEM2, SIEM4, and SIEM6. The figure shows stable discharge plateaus across the three ionogels, but their capacities significantly differ. The order is as follows: SIEM2 (108.7 mAh/g) > SIEM4 (95.6 mAh/g) > SIEM6 (76.5 mAh/g). This outcome can be attributed to factors such as lower interfacial impedance and reduced activation energy, which contribute to the superior discharge capacity of SIEM2. As the concentration of MTES increases, the interface impedance (R_i_) is progressively amplified, thereby degrading the ionic conductivity within the ionogel electrolyte and directly impacting the discharge efficiency.

The primary advantages of ionic liquids are their negligible vapor pressure and thermal stability compared to traditional solvents. This implies that they can offer a safer alternative in terms of flammability and volatility. However, the selection of cation–anion combinations is critically essential. In the highly reductive environment of lithium-ion batteries, the proton at the C2 position of the BMIM cation is the part of the structure most susceptible to reduction. Upon reduction, the BMIM structure may decompose, potentially affecting the performance and stability of the ionic liquid. As shown in Appendix A, as the number of cycles increases, the capacity gradually decreases, which may be the main reason for the poor charge–discharge cycling efficiency of the battery design we have proposed.

## 4. Conclusions

This study successfully immobilized ionic liquids in a silicon dioxide structure and utilized methyltriethoxysilane (MTES) to enhance the mechanical strength of silicon dioxide ionogel electrolytes. Through FT-IR spectroscopic analysis, we could observe the appearance of the absorption peak of ≡Si-O-Si≡ in the range of 1000~1200 cm^−1^ and the characteristic absorption peak of silane at 1275 cm^−1^. This evidence indicated the successful hydrolysis–condensation reaction of TEOS and MTES, forming a silicon dioxide structure. DSC results further demonstrated the confinement effect of ionic liquids. When the ionic liquid was confined within the silicon dioxide structure, its glass transition temperature (Tg) increased. With the addition of MTES, the mechanical properties of the ionogel electrolyte can be improved, simplifying its preparation.

It is worth noting that the introduction of silicon dioxide resulted in an activation energy lower than that of the ionic liquid, proving that incorporating silicon dioxide can promote the conduction of lithium ions. The linear sweep voltammetry (LSV) results showed that this novel silicon dioxide ionogel electrolyte could withstand voltages above 5 V without undergoing oxidative decomposition. Under a discharge rate of 0.2 C and conditions of 55°C, discharge capacity tests showed that the test batteries composed of silica ionogel electrolyte could achieve a discharge capacity of up to 108.7 mAh/g after the first discharge–charge cycle, providing an empirical evaluation of the performance of lithium solid-state batteries.

This study successfully prepared silicon dioxide ionogel electrolytes, immobilizing ionic liquids in a silicon dioxide structure. This novel electrolyte exhibited excellent ion conductivity and electrochemical properties and was successfully applied in lithium batteries. The findings of this research reveal significant potential and promise. In the future, we will contemplate employing impedance spectroscopy with non-blocking electrodes to measure Li⁺ transference number (T_Li⁺). This method allows for the direct probing of the mobility of ionic species without the intricacies of electrode reactions, offering a straightforward assessment of how the ionogel structure impacts the mobility of Li⁺ relative to other ions. Given the inherent challenges of lithium salt ILs, enhancements in the ionogel matrix and T_Li⁺ might pave the way for developing next-generation electrolytes and more promising lithium secondary batteries.

## Data Availability

The data that support the findings of this study are available from the corresponding author upon request.

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
