# Peer review of "Preparation and Characterization of Silica-Based Ionogel Electrolytes and Their Application in Solid-State Lithium Batteries"

_polymers, 2023, doi:10.3390/polym15173505_

Round 1

Reviewer 1 Report

Authors must identify the acronym 3C products (as computer, communication, and consumer electronic products) for better understanding. Authors must explain why MTES/TEOS ratios were prepared: 0, 0.2, 0.4, 0.6, 0.8, and 1.0.

Fine revision of English

Author Response

First, we are very grateful to the Reviewer for his review and recognition of this paper. Regarding your suggestions for additions and clarifications, our responses are as follows:

  • We have added “3C products (such as computer, communication, and consumer electronic products)” in the second paragraph of the Introduction section.
  • Regarding the MTES/TEOS ratio, we have provided additional clarification in Section 2.4 about the purpose of adding MTES. The objective is to enhance the mechanical strength of the silica ionogel electrolyte. The added content is as follows:

"Due to the gel-like consistency of BMIM-ClO4, the silica ionogel electrolyte (SIE) obtained from its hydrolytic condensation with TEOS exhibits suboptimal mechanical strength. When fabricated into thin films, this results in surface cracking. To improve its processability and yield a film with enhanced mechanical properties, the proportion of MTES can be increased. For instance, with a molar ratio 1.0, the design is based on MTES/ TEOS with 1.0 mole of MTES and 1.0 mole of TEOS. This leads to the formulation of a silica composite substrate. In this study, the substrate formulations were controlled at molar ratios of 0.2, 0.4, 0.6, 0.8, and 1.0. We synthesized a series of SiO2 ionogel polymer electrolytes and conducted identification and property analyses. The formulation and code of each sample are detailed in Table 1."

Reviewer 2 Report

The authors present a timely study on the use of ionogels for lithium batteries. It is an interesting topic, and an enjoyable read.

Unfortunately there are two major points that must be addressed before further consideration of this manuscript.

 (i) The following references appear to be non-existent articles: 12, 13, 15, 16, 17, 19, 20, 21, 23, 24, 29 and 31.

 (ii) Critical experimental details are not provided in the Experimental section:

*full descriptions of analytical methods

*full descriptions of electrochemical measurements

If the authors can amend these issues, it will be possible to thoroughly review the paper. I would be happy to see a revised edition of the manuscript.

Thank you

Author Response

First and foremost, we express our deepest gratitude to the reviewer for their thorough evaluation and positive feedback on our manuscript. Regarding the suggestions for additional clarification and elaboration, our responses are as follows:

  • Firstly, you mentioned that some references refer to non-existent articles. After a meticulous review, we realized that the errors arose while consolidating our manuscript. We greatly appreciate your identifying this significant oversight during the inspection, which has prevented potential confusion for future readers. We have now rechecked, removed the erroneous articles (References 12~29), and cited the correct literature with proper attribution. The corrected references are as follows:

  1. Ahmad, S.; Deepa, M. Ionogels encompassing ionic liquid with liquid like performance preferable for fast solid state electrochromic devices. Electrochemistry Communications 2007, 9, 1635-1638.
  2. Kim, J.-D.; Mori, T.; Kudo, T.; Honma, I. Ionogel electrolytes at medium temperatures by composite of ionic liquids with proton conducting cesium hydrogen sulfate. Solid State Ionics 2008, 179, 1178-1181.
  3. Mizumo, T. Rubbery Ionogels and Glassy Ionogels: Design and Ion Conduction. Kobunshi Ronbunshu 2008, 65, 516-524.
  4. Echelmeyer, T.; Meyer, H. W.; van Wüllen, L. Novel ternary composite electrolytes: Li ion conducting ionic liquids in silica glass. Chemistry of Materials 2009, 21, 2280-2285.
  5. Petit, D.; Korb, J. P.; Levitz, P.; LeBideau, J.; Brevet, D. Multiscale dynamics of 1H and 19F in confined ionogels for lithium batteries. Comptes Rendus Chimie 2010, 13, 409-411.
  6. Gayet, F.; Viau, L.; Leroux, F.; Monge, S.; Robin, J.-J.; Vioux, A. Polymer nanocomposite ionogels, high-performance electrolyte membranes. Journal of Materials Chemistry 2010, 20, 9456.
  7. Le Bideau, J.; Viau, L.; Vioux, A. Ionogels, ionic liquid based hybrid materials. Chem Soc Rev 2011, 40, 907-925.
  8. Wu, F.; Tan, G.; Chen, R.; Li, L.; Xiang, J.; Zheng, Y. Novel Solid-State Li/LiFePO4 Battery Configuration with a Ternary Nanocomposite Electrolyte for Practical Applications. Advanced Materials 2011, 23, 5081-5085.
  9. Le Bideau, J.; Ducros, J.-B.; Soudan, P.; Guyomard, D. Solid-State Electrode Materials with Ionic-Liquid Properties for Energy Storage: the Lithium Solid-State Ionic-Liquid Concept. Advanced Functional Materials 2011, 21, 4073-4078.
  10. Xie, Z.-L.; Xu, H.-B.; Geßner, A.; Kumke, M. U.; Priebe, M.; Fromm, K. M.; Taubert, A. A transparent, flexible, ion conductive, and luminescent PMMA ionogel based on a Pt/Eu bimetallic complex and the ionic liquid [Bmim][N(Tf)2]. Journal of Materials Chemistry 2012, 22, 8110.
  11. Noor, S. A. M.; Bayley, P. M.; Forsyth, M.; MacFarlane, D. R. Ionogels based on ionic liquids as potential highly conductive solid state electrolytes. Electrochimica Acta 2013, 91, 219-226.
  12. Guyomard-Lack, A.; Delannoy, P. E.; Dupre, N.; Cerclier, C. V.; Humbert, B.; Le Bideau, J. Destructuring ionic liquids in ionogels: enhanced fragility for solid devices. Phys Chem Chem Phys 2014, 16, 23639-23645.
  13. Horowitz, A. I.; Panzer, M. J. Poly(dimethylsiloxane)-Supported Ionogels with a High Ionic Liquid Loading. Angew Chem Int Ed Engl 2014, 53, 9780-9783.
  14. Nordstrom, J.; Aguilera, L.; Matic, A. Effect of Lithium Salt on the Stability of Dispersions of Fumed Silica in the Ionic Liquid BMImBF4. Langmuir 2012, 28, 4080-4085.
  15. Makita, K.; Akamatsu, Y.; Yamazaki, S.; Kai, Y.; Abe, Y. Surface Morphology of Silica Films Derived by the Sol-Gel Method and Its Application to a Water Repellent Glass. Journal of the Ceramic Society of Japan 1997, 105(1227), 1012-1017.
  16. Legrand, A. P.; Hommel, H.; Tuel, A.; Vidal, A.; Balard, H.; Papirer, E.; Levitz, P.; Czernichowski, M.; Erre, R.; Van Damme, H.; Gallas, J. P.; Hemidy, J. F.; Lavalley, J. C.; Barres, O.; Burneau, A.; Grillet, Y. Hydroxyls of silica powdersHydroxyles des silices divisees. Advances in Colloid and Interface Science 1990, 33, 91-330.
  17. Alkhadra, M. A.; Su, X.; Suss, M. E.; Tian, H.; Guyes, E. N.; Shocron, A. N.; Conforti, K. M.; de Souza, J. P.; Kim, N.; Tedesco, M.; Khoiruddin, K.; Wenten, I. G.; Santiago, J. G.; Hatton, T. A.; Bazant, M. Z. Electrochemical Methods for Water Purification, Ion Separations, and Energy Conversion. Chemical Reviews 2022, 122 (16), 13547-13635.
  18. Bard, A. J.; Faulkner, L. R. "Electrochemical Methods: Fundamentals and Applications" (2nd Edition), 2000.

  • Regarding the analytical methods and electrochemical measurements in the experimental section, there was a lack of clear description, which is an oversight on our part. Consequently, we have added supplementary explanations in sections 2.5 and 2.6 of the manuscript, as follows:

2.5 Materials Identification and Analysis Methods

    In this study, Fourier-transform infrared (FTIR) spectroscopy was utilized to identify the characteristic peaks of ionic liquids. Experimental conditions included taking an appropriate amount of the sample and blending it with KBr powder in a 1:100 ratio. This mixture was then finely ground and pressed to form a KBr pellet for spectroscopic analysis. The spectral resolution was 4 cm-1 with a scanning wavenumber range of 4000 to 400 cm-1. Differential scanning calorimetry (DSC) was employed to determine the glass transition temperature (Tg) of the silica-based ionogel electrolyte (SIE). This measurement provided insights into the relationship of its ionic conduction within a specific temperature range. The sample mass for testing ranged from 5 to 10 mg. Measurements were conducted under a nitrogen atmosphere with a heating rate of 10°C/min, spanning a temperature range of -120°C to 30°C.

   Time-evolution 1H-NMR spectra of the as-prepared BMIM-ClO4 were characterized by a Bruker Avance Neo 9.4 Tesla NMR spectrometer (corresponding to a proton resonance frequency of approximately 400.13 MHz) equipped with a liquid iProbe. Analysis of the SIE using 29Si-NMR spectra was conducted with a Bruker 400 WB solid-state NMR spectrometer, operating at a proton frequency of 400.13 MHz. The NMR spectra were acquired under the following conditions: 29Si frequency of 79.48 MHz, a π/2 pulse of 3.9 µs. The single pulse sequence included a π/4 pulse of 3.9 µs, a decoupling duration of 6.3 µs, and a recycle delay of 100 seconds, totaling 2000 scans. Samples were packed into 4 mm zirconia rotors and spun at 9 kHz under an airflow.

2.6 Descriptions of electrochemical measurements

    Ionic conductivities were assessed in the range of 30~100℃ within a vacuum chamber, employing an HP4192A impedance analyzer (Agilent) operating over frequencies from 0.1 Hz to 1 MHz. The silica ionogel electrolytes were positioned between two Au electrodes. The vacuum within the chamber was maintained at a pressure of 5×10-2 torr during the Nyquist impedance measurements.

    Linear sweep voltammetry (LSV) experiments were conducted using a CR2032 coin cell. The cell configuration involved stainless steel as the working electrode, metallic lithium as the counter electrode, and silica ionogel electrolyte interposed between them, forming a half-cell assembly. The entire assembly process was meticulously executed within a dry box (Model: UNI-LAB), ensuring an environment devoid of water and oxygen. The potentiostatic/galvanostatic measurements were facilitated by an AUTOLAB PGSTAT30 instrument, with the voltammetric scan range set from 0 to 7V and a scan rate of 5mV/s. This setup enabled a detailed examination of the current response as a function of voltage.

    For charge-discharge experiments, metallic lithium was employed as the anode, and LiFePO4 was chosen as the cathode (with a theoretical capacity of 170 mAh/g). The ionogel electrolyte developed in this study was sandwiched between the anode and cathode, forming a CR2032 lithium half-cell. The assembly for this configuration was strictly carried out in a dry box environment. The charge and discharge assessments were performed using the LAND CT2100A system at a rate of 0.2C. Furthermore, the cell was subjected to a thermal environment 55℃ to evaluate its charge-discharge performance.

Round 2

Reviewer 2 Report

Summary:

The authors present a timely study on the use of ionogels for lithium batteries. It is an important and interesting topic, will be of interest to a broad readership, and an enjoyable read. The concept is a worthwhile and suitable application. The work is also well-placed in terms of being up to date with most of the recent relevant literature.

The manuscript is in general very well written, well structured, and some parts of the work are scientifically sound. Although there is originality in the combination of the specific IL and silica used in this work, the significance is moderate – the work is not ground-breaking in my opinion.

Considering the manuscript section-by-section, the abstract is well-written and suitable, the introduction describes well the current state-of-the-art, the experimental section is reasonably well detailed (some omissions, see below), and the results and discussion section is logically arranged, but with some issues (see below). The conclusions are adequate.

Whilst the manuscript appears to be in general fairly sound, there are several important points/concerns that I feel need addressing before the scientific rigour of the work can be confirmed. These are summarized below. I would be pleased to receive and consider the authors response to these points.

1. On page 2.

“The widespread attention ionogels have received is mainly due to their characteristics: non-volatility, non-flammability, high ionic conductivity, and wide electrochemical windows.”

 Of these, none (or perhaps only high ionic conductivity) are inherent characteristics of ionogels. Indeed, there are many ionic liquids that have volatility, flammability and poor electrochemical windows, and so any ionogel comprised of such ionic liquids would also have similar properties. I suggest amending this sentence to something like:

“The widespread attention ionogels have received is mainly due to their characteristics (with suitable choices of components): low volatility, low flammability, high ionic conductivity, and wide electrochemical windows.”

2. On page 2 (and there-after).

The authors use various different abbreviations for the bis(trifluoromethanesulfonyl)amide anion. Sometimes TFSI, sometimes NTf2, and even N(Tf)2. Please choose one abbreviation and use it throughout (also be sure to define it).

3. Omissions from the experimental section.

In the “2.1. Chemical and materials” sub-section:

source of LFP (and any conductive carbon/binder etc)

In the “2.5. Materials Identification and Analysis Methods” sub-section:

DSC: the full temperature program e.g. the starting temperature and rate of cooling down to −120 °C are not described, also, is it just one cooling heating cycle, or second cycle? Please provide the full details.

NMR: “Time evolution” is mentioned, but there are no time evolution data provided – is this an error, or does it mean the NMR spectra were periodically measured until the spectra showed that the reaction was complete?  Were the NMR spectra of the IL measured in a single tube or double tube arrangement, what was the internal/external reference/solvent? What was the temperature? Please provide the full details.

In the “2.6. Descriptions of electrochemical measurements” sub-section:

Ionic conductivity: what was the amplitude (mV rms)? Diameter of electrodes? Diameter and thickness of electrolyte? Any other separator?

LSV: what was the temperature? Diameter of electrodes? Diameter and thickness of electrolyte? Any other separator?

Charge/discharge: please provide full details of the LFP electrode (including its composition of LFP/conductive carbon/binder, and if it was purchased or prepared (with full description of preparation)? Diameter of electrodes? Diameter and thickness of electrolyte? Any other separator?

4. Page 8, description of successful synthesis of IL.

Whilst the authors provide NMR and FTIR data to support their assumption that they have successfully synthesised a pure sample of BMIMClO4, the FTIR and NMR data do not provide any means to check for complete ion exchange Cl to ClO4 . The presence of ClO4 is clear from the FTIR, but is there any residual Cl? Some other techniques may be necessary.

4. Page 10 and Figure 5.

Please provide a table summarising the Tg of each sample, or write them on the chart. What is the feature in the curve for BMIMClO4 at around −40 °C?

4. Page 10.

“confirming that the ionic liquid is indeed encapsulated in the silica scaffold, referred to as the confinement effect”

It would be best to include a suitable reference (citation) that describes the confinement effect in detail. Also, is the slight shift in Tg really sufficient evidence to make the claim of encapsulation?

5. Page 15 and Figure 9.

Determining the oxidative decomposition limit of the electrolyte by LSV is appropriate. However, in the procedure used by the authors, they scan the range between 0V and 7V. I can only assume that the potential is initially stepped down from the resting potential (open circuit potential, OCP) to 0 V, and then the voltage sweep begins. This is a highly unusual (and not recommended) way to perform this experiment, as the potential is initially driven into the lithium deposition/stripping range, and there may be some reaction at the electrode surface which will influence the behaviour observed when the potential subsequently reaches the high potential range e.g. 5 V. A more usual procedure would be to simply sweep up from the OCP (or a potential close to the OCP). The reductive range would typically be measured in a separate experiment (with suitable electrodes), sweeping down from the OCP (or a potential close to the OCP). Can the authors provide some explanation for why they did it this way (i.e. dropping down to 0V then sweep all the way up)?

The data beyond ~5.5 V also seem unnecessary. It is quite likely that the stainless steel electrode is also not stable at such high voltages, and so the data are of little consequence I think.

6. Page 17 and Figure 11.

“This is due to the surface becoming rougher as the methyl content increases, as observed in the SEM surface morphology image in Figure 11”.

It is not clear from the SEM micrographs provided that the surface is substantially “rougher” with increasing methyl content. Can the authors provide some further description/explanation?

7. Page 18 and Figure 12

Please provide the first cycle data (charts), and also some data showing the decay in capacity with cycle number for the electrolytes shown in Figure 12 (as supporting information would be fine).

8. Page 18

“A small amount of water in the electrolyte causes the formation of a LiOH passivation layer during the lithium ions' charging and discharging process”

Having a low water content is critical for lithium ion batteries. Whils the authors mention in the experimental section that the electrolytes are prepared in rigourously dry conditions, here they mention water content. Water content is concerning as not only will it hinder the reversible reactions that are intended to occur in the battery, but also can significantly influence the measured values for conductivity etc. Since the imidazolium based ILs are quite hydrophilic (as is silica of course), itr is a major concern for the future application of this type of electrolyte. Can the authors discuss this?

9. A comment on the IL

The specific IL being used here, BMIMClO4 does not strike me as a particular promising choice of IL for this application.. It is well known that imidazolium cations exhibit reductive instability in Li-ion batteries (due to reactivity of the C2 proton). Also, the perchlorate anion is a strong oxidiser and shows some thermal instability. Do the authors have some comment?

10. A suggestion

In future work it would be of interest to explore the Li+ transference number (this can be measured quite simple\y by impedance measurement with non-blocking electrodes). Since ILs with lithium salts usually exhibit very poor Li+ transference numbers, it would be very interesting to know if this is improved in the ionogel.

Thank you.

Author Response

Firstly, we would like to express our sincere gratitude to the Reviewer for the second round of review and affirmation of our manuscript. We also appreciate the insights and queries that were raised, for which we have endeavored to clarify and provide explanations. With respect to your suggestions that require additional information and clarification, we have marked our responses in blue text, as follows:

  1. On page 2.

“The widespread attention ionogels have received is mainly due to their characteristics: non-volatility, non-flammability, high ionic conductivity, and wide electrochemical windows.”

Reply : It has been revised to the following sentence. Thank you for your correction. “The widespread attention ionogels have received is mainly due to their characteristics (with suitable choices of components): low volatility, low flammability, high ionic conductivity, and wide electrochemical windows.”

  1. On page 2 (and there-after).

The authors use various different abbreviations for the bis(trifluoromethanesulfonyl)amide anion. Sometimes TFSI, sometimes NTf2, and even N(Tf)2. Please choose one abbreviation and use it throughout (also be sure to define it).

Reply : This is because different authors in various references use different abbreviations for "bis(trifluoromethanesulfonyl)amide anion." However, we have now standardized and are using "TFSI" as the abbreviation. Thank you for bringing this to our attention.

  1. Omissions from the experimental section.

In the “2.1. Chemical and materials” sub-section:

source of LFP (and any conductive carbon/binder etc)

Reply : Regarding the binder and conductive carbon materials used in the fabrication of the LFP cathode, we have provided additional information in the “2.1. Chemical and materials” sub-section. The added content is as follows:

Super P (high conductivity carbon black) (UBIQ Technology Co. Ltd., Taipei, Taiwan); Poly(1,1-difluoroethylene) (PVDF) (UBIQ Technology Co. Ltd., LTD., Taipei, Taiwan); 1-methyl-2-pyrrolidone (NMP, C5H9NO) (99%, Thermo Fisher Scientific Inc., Waltham, MA, USA).

In the “2.5. Materials Identification and Analysis Methods” sub-section:

DSC: the full temperature program e.g. the starting temperature and rate of cooling down to −120 °C are not described, also, is it just one cooling heating cycle, or second cycle? Please provide the full details.

Reply :  We have provided additional information in the “2.5. Materials Identification and Analysis Methods” sub-section. The added content is as follows:
   Samples were evaluated with a mass ranging between 5~10 mg. Experiments were executed under a nitrogen atmosphere. The initial heating profile spanned from -120°C to 120°C at a controlled ramp rate of 10°C/min. Following this, the sample underwent a cooling phase, descending at a rate of 10°C/min until reaching -120°C. Subsequently, a secondary heating cycle was administered, elevating the temperature from -120°C to 30°C at the same rate of 10°C/min. During this second thermal excursion, the glass transition temperature (Tg) of the material was closely monitored and documented.

NMR: “Time evolution” is mentioned, but there are no time evolution data provided – is this an error, or does it mean the NMR spectra were periodically measured until the spectra showed that the reaction was complete?  Were the NMR spectra of the IL measured in a single tube or double tube arrangement, what was the internal/external reference/solvent? What was the temperature? Please provide the full details.

Reply :

  1. In the section describing the 1H-NMR measurement, we have deleted the term “Time evolution”. Thank you for pointing it out.
  2. The sample was loaded into a single tube for analysis, utilizing D2O as the deuterated solvent. Trace amounts of H2O, resonating at δ=4.7 ppm, were employed as the internal reference. The measurements were conducted at a maintained temperature of 25°C.

Solid-state 29Si-NMR : Q8M8(octamethylcyclotetrasiloxane) were used as external secondary references. The measurements were conducted at a maintained temperature of 25°C. 

In the “2.6. Descriptions of electrochemical measurements” sub-section:

Ionic conductivity: what was the amplitude (mV rms)? Diameter of electrodes? Diameter and thickness of electrolyte? Any other separator?

Reply : We have provided additional information in the “2.6. Descriptions of electrochemical measurements” sub-section. The added content is as follows:
   Ionic conductivity measurements were set at an amplitude of 50 mV rms. The diameter of the measuring electrode was 12 mm, while the electrolytes had a diameter of 15 mm and a thickness maintained at 100 μm. No separators were incorporated.

LSV: what was the temperature? Diameter of electrodes? Diameter and thickness of electrolyte? Any other separator?

Reply : The LSV experiments were conducted utilizing a CR2032 coin cell. Within this cell, stainless steel, with a diameter of 12mm, served as the working electrode, while metallic lithium of the same diameter acted as the counter electrode. Positioned between these electrodes was a silica ionogel electrolyte with a diameter of 15mm and a controlled thickness of 100μm. Notably, no additional separators were employed in this half-cell configuration. The assembly of the half-cell was meticulously carried out within a dry box (Model: UNI-LAB) to ensure that the humidity and oxygen levels remained below 1 ppm. Measurements were facilitated by a Potentiostat/galvanostatic instrument (Model: AUTOLAB PGSTAT30), with a voltammetric scan range set from 0 to 7V and a scan rate of 5mV/s, under a controlled temperature of 30°C.

Charge/discharge: please provide full details of the LFP electrode (including its composition of LFP/conductive carbon/binder, and if it was purchased or prepared (with full description of preparation)? Diameter of electrodes? Diameter and thickness of electrolyte? Any other separator?

Reply : 1.0 g of anhydrous Super P (a highly conductive carbon black) and 1.0 g of a 4.0 wt% PVDF solution in NMP were combined in a 50 ml sample vial. This mixture was stirred at 200 rpm for an hour to ensure uniform blending. Subsequently, 8.0 g of LiFePO4 was added and stirred again at 200 rpm for another hour. Upon achieving homogeneity, an additional 10 ml of NMP was incorporated, and the resulting slurry was stirred at 500 rpm for 4 hours. This uniform paste was then blade-coated onto an aluminum foil substrate (paste thickness: 300μm; foil thickness: 20μm). The coated foil was dried overnight at 80℃ under vacuum conditions. Once dried, the electrode material was punched into 13mm diameter discs. The final weight of each electrode disc was measured with an electronic balance to determine the loading of LiFePO4. The ionogel electrolyte prepared in this study was sandwiched between the anode and cathode, with a diameter of 15 mm and thickness maintained at 100 μm. Notably, no additional separators were employed within the cell.

  1. Page 8, description of successful synthesis of IL.

Whilst the authors provide NMR and FTIR data to support their assumption that they have successfully synthesised a pure sample of BMIMClO4, the FTIR and NMR data do not provide any means to check for complete ion exchange Cl to ClO4 . The presence of ClO4 is clear from the FTIR, but is there any residual Cl? Some other techniques may be necessary.

Reply : To prepare the BMIM-ClO4 ionic liquid, dissolve the synthesized BMIC (1-butyl-3-methylimidazolium chloride) in a predetermined amount of sodium perchlorate (NaClO4, with a 1:1 molar ratio). Use acetone as the solvent and mechanically stir the mixture at room temperature for 7 days. At this point, BMIM-ClO4 ionic liquid and sodium chloride precipitate will form. Use vacuum filtration to remove the sodium chloride precipitate and a vacuum concentrator to remove the acetone. Finally, use a freeze dryer to remove any remaining impurities to obtain the BMIM-ClO4(1-butyl-3-methylimidazole perchlorate) ionic liquid with the chloride ions replaced by perchlorate ions. Next, purify the obtained BMIM-ClO4 ionic liquid using activated carbon. Mix the BMIM-ClO4 with activated carbon (activated carbon weight = BMIM-ClO4 weight × 0.2) and use ethanol as the solvent. Maintain the temperature at 70°C and purify the mixture for 7 days. Then, use vacuum filtration to remove the activated carbon and a vacuum concentrator to remove the ethanol. Finally, use a freeze dryer to remove any remaining impurities to obtain the purified BMIM-ClO4 ionic liquid.

    Based on the synthesis process of BMIM-ClO₄ ionic liquid and the subsequent rigorous purification steps, we believe that if there is any residual chloride ion present in BMIM-ClO₄, it would be in extremely trace amounts. With the instruments currently available in our laboratory, detecting such minute levels of chloride ions would be challenging. If we were to employ advanced X-Ray fluorescence spectroscopy for detection, finding a research center to assist us on short notice would be difficult, as reservations for such instruments are typically required well in advance.

  1. Page 10 and Figure 5.

Please provide a table summarising the Tg of each sample, or write them on the chart. What is the feature in the curve for BMIMClO4 at around −40 °C?

Reply : Added Table 2 Summary of the Tg and Tc for each sample.

Table 2  Summary of the glass transition temperature (Tg) and crystallization temperature (Tc) for each sample.

Sample

Tg(℃)

Tc(℃)

BMIM-ClO4

-72.2

-40.5

BMIM-ClO4 + LiClO4

-64.2

--

SIEM0

-62.8

--

SIEM2

-80.8

--

SIEM4

-63.3

--

SIEM6

-63.5

--

SIEM8

-64.3

--

SIEM10

-62.8

--

In the thermal profile of BMIM-ClO₄, besides the observed Tg at -72.2℃, an exothermic peak(Tc) emerged at -40.5℃. It is postulated that BMIM-ClO₄, having undergone a comprehensive heating and cooling cycle, exhibits a more ordered molecular arrangement. Strong electrostatic interactions might be present within BMIM-ClO₄, leading to certain crystallization events, thus accounting for the appearance of the exothermic peak.

  1. Page 10.

“confirming that the ionic liquid is indeed encapsulated in the silica scaffold, referred to as the confinement effect”

It would be best to include a suitable reference (citation) that describes the confinement effect in detail. Also, is the slight shift in Tg really sufficient evidence to make the claim of encapsulation?

Reply : "Confinement effects" refer to the phenomena wherein the physical, chemical, or biological characteristics of a substance (such as molecules, ions, particles, etc.) exhibit behaviors different from those in an unrestricted environment when confined within a relatively small space or volume. Such effects are prevalent in nanostructures, thin films, and pores on the micro or nanometric scale. Due to this spatial confinement, the properties of the material are influenced by its environment, leading to potential changes in its kinetics, electrochemistry, thermal properties, and mechanical attributes[28,29].

    The confinement effects in silica ionogel electrolytes focus on the property changes of BMIM-ClO4 when spatially restricted within a silica scaffold. Ion liquids confined within a networked structure may exhibit alterations in their physical properties, such as their glass transition temperature (Tg), due to the constrained mobility of the ionic liquid. Indeed, there is a significant decrease when comparing the Tg of SIEM2 (-80.8°C) with that of BMIM-ClO4+LiClO4 (-64.2°C). The Tg variations for other ionogel electrolytes might be minor, hypothesizing that a more stable phase equilibrium state is established between the silica networked structure and the ionic liquid[30]. However, from the SEM observations of the ionogel electrolyte, we can ascertain that the ionic liquid is indeed encapsulated within the silica scaffold.

    We also cite three papers to substantiate the confinement effects, elucidating the variations in the physical and chemical characteristics of Ionic Liquids or liquid substances within mesoporous materials.

[28] Huber, P. Soft matter in hard confinement: phase transition thermodynamics, structure, texture, diffusion and flow in nanoporous media. J. Phys.: Condens. Matter 2015, 27, 103102.

[29] Abdou, N.; Alonso, B.; Brun, N.; Devautour-Vinot, S.; Paillet, M.; Landois, P.; Mehdi, A.; Hesemann, P. Confinement effects on the ionic liquid dynamics in ionosilica ionogels: impact of the ionosilica nature and the host/guest ratio. J. Phys. Chem. C 2022, 126, 49, 20937–20945.

[30] Alba-Simionesco, C.; Coasne, B.; Dosseh, G.; Dudziak, G.; Gubbins, K. E.; Radhakrishnan, R.; Sliwinska-Bartkowiak, M. Effects of confinement on freezing and melting. J. Phys.: Condens. Matter 2006, 18, R15–R68.

  1. Page 15 and Figure 9.

Determining the oxidative decomposition limit of the electrolyte by LSV is appropriate. However, in the procedure used by the authors, they scan the range between 0V and 7V. I can only assume that the potential is initially stepped down from the resting potential (open circuit potential, OCP) to 0 V, and then the voltage sweep begins. This is a highly unusual (and not recommended) way to perform this experiment, as the potential is initially driven into the lithium deposition/stripping range, and there may be some reaction at the electrode surface which will influence the behaviour observed when the potential subsequently reaches the high potential range e.g. 5 V. A more usual procedure would be to simply sweep up from the OCP (or a potential close to the OCP). The reductive range would typically be measured in a separate experiment (with suitable electrodes), sweeping down from the OCP (or a potential close to the OCP). Can the authors provide some explanation for why they did it this way (i.e. dropping down to 0V then sweep all the way up)?

The data beyond ~5.5 V also seem unnecessary. It is quite likely that the stainless steel electrode is also not stable at such high voltages, and so the data are of little consequence I think.

Reply : In our latest research endeavor, we have explored the application of ionogel electrolytes in the field of lithium-ion batteries, a novel topic for our team. Consequently, we adopted some innovative testing methodologies that might deviate from standard international protocols. While our multiple test results suggest that these methods might not have negative implications on the electrode surface, we genuinely appreciate the reviewer's insights. Moving forward, our LSV tests will be configured within a range from OCP to 5V, undergoing repeated cyclic scans to observe the oxidation-reduction phenomena of the electrolytes.

  1. Page 17 and Figure 11.

“This is due to the surface becoming rougher as the methyl content increases, as observed in the SEM surface morphology image in Figure 11”.

It is not clear from the SEM micrographs provided that the surface is substantially “rougher” with increasing methyl content. Can the authors provide some further description/explanation?

Reply : In Figure 11, we have incorporated AFM (Atomic Force Microscopy) images that scan the surface of the ionogel electrolytes. These visuals underscore the increasing surface roughness or irregularities, accompanied by minute indentations, as the content of methyl (MTES) escalates.

Figure 11. The SEM and AFM images of the silicate ion gel electrolyte. SEM images: (a) SIEM2 (b) SIEM4 (c) SIEM6 (d) SIEM8, AFM images: (e) SIEM2 (f) SIEM4 (g) SIEM6 (h) SIEM8

  1. Page 18 and Figure 12

Please provide the first cycle data (charts), and also some data showing the decay in capacity with cycle number for the electrolytes shown in Figure 12 (as supporting information would be fine).

Reply : We have added the following description in the ” Section 3.3. Charge and discharge test of solid-state lithium batteries”.

Figure 12 shows stable discharge plateaus across the three ionogels, yet their capacities significantly differ. The order is as follows: SIEM2 (108.7 mAh/g) > SIEM4 (95.6 mAh/g) > SIEM6 (76.5 mAh/g). This outcome can be attributed to factors such as lower interfacial impedance and reduced activation energy, which contribute to the superior discharge capacity of SIEM2. As the concentration of MTES increases, the interface impedance (Ri) progressively amplifies, thereby deteriorating the ionic conductivity within the ionogel electrolyte and directly impacting the discharge efficiency.

  The primary advantages of ionic liquids are their negligible vapor pressure and thermal stability compared to traditional solvents. This implies that they can offer a safer alternative in terms of flammability and volatility. However, the selection of cation-anion combinations is critically essential. In the highly reductive environment of lithium-ion batteries, the proton at the C2 position of the BMIM cation is the most susceptible part of the structure to reduction. Upon reduction, the BMIM structure may decompose, potentially affecting the performance and stability of the ionic liquid. As shown in Table SI, as the number of cycles increases, the capacity gradually decreases, which may be the main reason for the poor charge-discharge cycling efficiency of the battery design we have proposed.

Please refer to the supporting information(Table SI).

Table SI  Cyclic charging and discharging test (discharge capacity) of 2032 coin cell assembled with ionogels electrolyte.

Sample code

Cycle 1

Cycle 5

Cycle 10

(mAh/g)

(mAh/g)

(mAh/g)

SIEM2

108.7

77.1

43.5

SIEM4

95.6

65.0

32.5

SIEM6

76.5

54.5

25.2

  1. Page 18

“A small amount of water in the electrolyte causes the formation of a LiOH passivation layer during the lithium ions' charging and discharging process”

Having a low water content is critical for lithium ion batteries. Whils the authors mention in the experimental section that the electrolytes are prepared in rigourously dry conditions, here they mention water content. Water content is concerning as not only will it hinder the reversible reactions that are intended to occur in the battery, but also can significantly influence the measured values for conductivity etc. Since the imidazolium based ILs are quite hydrophilic (as is silica of course), itr is a major concern for the future application of this type of electrolyte. Can the authors discuss this?

Reply : To prevent any potential misinterpretation by the readers, we have removed the last paragraph in Section 3.3 that stated, “A small amount of water in the electrolyte leads to the formation of a LiOH passivation layer during the lithium ions' charge-discharge cycle.” Further clarification has been added to the second paragraph of Section 3.3 as follows:

    In this study, the ionogel electrolytes underwent a vacuum freeze-drying process to ensure the complete removal of water content. The presence of water not only obstructs the intended reversible reactions within the lithium battery but also could significantly influence the conductivity and ionic transport of the material, subsequently diminishing the battery's efficiency. Given that BMIM-ClO4 is inherently hydrophilic and the structure of silica also exhibits hydrophilic tendencies, all components and materials for lithium battery assembly underwent rigorous dehydration. They were stored in a dry box until they were fully assembled into coin cells, only after which subsequent battery charge-discharge tests were conducted.

  1. A comment on the IL

The specific IL being used here, BMIMClO4 does not strike me as a particular promising choice of IL for this application.. It is well known that imidazolium cations exhibit reductive instability in Li-ion batteries (due to reactivity of the C2 proton). Also, the perchlorate anion is a strong oxidiser and shows some thermal instability. Do the authors have some comment?

Reply : One of the primary advantages of ionic liquids is their negligible vapor pressure and thermal stability compared to conventional solvents. This implies that they offer a safer alternative concerning flammability and volatility. However, the choice of cation-anion pairing is paramount. In the highly reductive environment of lithium-ion batteries, the proton at the C2 position of the imidazolium cation is the most susceptible part of the structure to reduction. Upon its reduction, the imidazolium framework may decompose, which could influence the performance and stability of the ionic liquid. I surmise this to be a leading factor in the subpar charge-discharge cycling efficiency of the batteries we designed.

    Moreover, the use of ionic liquids containing a perchlorate anion in lithium-ion batteries does pose some potential safety concerns, particularly given the strong oxidizing nature of the perchlorate anion. Therefore, our future investigations on electrolyte materials will focus on other imidazolium salts paired with different anions, which may offer improved stability. Examples include BMIM-[PF6], BMIM-[TFSI], or N-methyl-N-propylpiperidinium bis(trifluoromethylsulfonyl)imide.

  1. A suggestion

In future work it would be of interest to explore the Li+ transference number (this can be measured quite simple by impedance measurement with non-blocking electrodes). Since ILs with lithium salts usually exhibit very poor Li+ transference numbers, it would be very interesting to know if this is improved in the ionogel.

Reply : Thank you for your insightful suggestions. In our future endeavors, we will opt for appropriate ionic liquids (ILs) and prioritize the research directions you recommended as one of the focal points in our subsequent studies. Investigating the Li⁺ transference number (T_Li⁺) in ionogels is indeed a pivotal next step in assessing their potential for battery applications. The structural design of ionogels, where an ionic liquid is embedded within a silicate matrix, could lead to distinct kinetics. The confinement within the ionogel matrix might mitigate the mobility disparities between Li⁺ ions and other anions. If the matrix selectively impedes the movement of larger anions, it could elevate the T_Li⁺.

    In the future, we will contemplate employing impedance spectroscopy with non-blocking electrodes to measure T_Li⁺. This method allows for the direct probing of the mobility of ionic species without the intricacies of electrode reactions, offering a straightforward assessment of how the ionogel structure impacts the mobility of Li⁺ relative to other ions. Given the inherent challenges of lithium salt ILs, enhancements in the ionogel matrix and T_Li⁺ might pave the way for the development of next-generation electrolytes and more promising lithium secondary batteries.

   In addition, we have combined Tables 4 and 5 into one table, facilitating easier comparison and reading.

Round 3

Reviewer 2 Report

Thank you for your thorough responses to my comments and queries.

I have just one remaining query/comment:

In the supporting information, the discharge data provided in the table for the 1st cycle appears to match the capacity reached for each of the three electrolytes shown in Figure 12 in the main manuscript. However, the caption of Fig 12 states "after the first charge-discharge cycle". Does Fig 12 show the first cycle discharge, or second cycle discharge? Please clarify, and please also include the corresponding charge curves (it is really important to see these, and to understand the behaviour). Also, since this is a LFP\Li half cell, rather than “first charge-discharge” cycle, it should be perhaps “first discharge-charge cycle”.

Author Response

We sincerely appreciate the reviewer's meticulous examination of our replies and details. In response to the questions and suggestions, we offer the following clarifications and corrections:

    Regarding lithium-ion batteries, the first charge-discharge cycle is typically considered the activation cycle for the cell after assembly. During this phase, the electrode material undergoes the initial lithium-ion intercalation and deintercalation processes, which might induce irreversible chemical reactions and microstructural changes. Consequently, the efficiency and capacity of the initial cycle might differ from subsequent cycles. Therefore, when evaluating the performance of a lithium-ion battery, results from the first cycle are often disregarded, with primary emphasis placed on the following cycles.

    Thus, the current Figure 12 presents the discharge-charge curve of the second cycle. Furthermore, we have amended the caption of Figure 12 to specify that it represents the second cycle's discharge-charge curve. Moreover, in Section 3.3 "Discharge and Charge Test of Solid-State Lithium Batteries" within the manuscript, all descriptions related to the charge-discharge have been revised to state 'discharge-charge' curves. We hope these revisions will render the experimental results more comprehensive.

Figure 12. Discharge-charge diagram of the second discharge-charge cycle of an assembled CR2032 button cell with lithium/silica ionogel electrolyte/LiFePO4.
